# Synthesis of bimetallic MOFs via interface control using gallium-based liquid metal

Jui-Chi Lin[1], Chun-Tse Wei[1], Chien-Hua Wang[2], Yu-Cheng Chang [1], Wen-Wei Wu[2] & Chun-Wei Huang [1] ✉

Metal-organic frameworks (MOFs) are porous crystalline materials whose composition and structure can be tuned for catalysis, sensing, and optoelectronic applications. Incorporating two different metal ions into a single framework can broaden the functionalities of MOFs. However, conventional solvothermal synthesis typically requires high temperature and long reaction times, while electrochemical anodic dissolution is limited by solid anodes that can only supply a single metal source. For active metals such as Mg, the rapid formation of a passivation layer on solid anodes hinders the continuous release of metal ions. Here, we introduce a liquid metal interface-controlled electrochemical strategy using a fluid Mg-Ga alloy anode, overcoming the single-metal limitation of conventional anodic deposition. The self-healing surface of liquid gallium and its high surface tension induce electrocapillarity and Marangoni flow under an applied bias, enabling dynamic and spatially uniform release of $Mg^{2+}$ ions that co-assemble precisely with $Zn^{2+}$ from solution. Under an optimized bias of 0.3 V, the resulting bimetallic ZnMg-MOF-74 exhibits high crystallinity, nanosheet morphology, and uniform metal distribution. The ZnMgO oxide derived from a two-step annealing process shows excellent ultraviolet photodetection performance, with a responsivity of 1.48 A/W and a fast rise time of 0.32 s. This substrate-free and composition-controllable platform highlights a strategy for interface-driven synthesis of multimetallic MOFs and their derivatives for sensing and optoelectronic applications.

Liquid metals have recently gained attention in materials science due to their unique combination of metallic conductivity and fluidic deformability. Among them, gallium (Ga) stands out for its low melting point (29.76 °C), high electrical conductivity ($7.88 \times 10^5$ S/m), strong surface tension, and excellent alloying ability[1–4]. Ga can remain stable in the liquid phase at room temperature and form homogeneous alloys with various metals[5–7]. More importantly, its self-healing interfaces and dynamic reconstruction behavior allow for stable and selective ion release in electrochemical reactions, enabling localized control over interfacial processes[8–11]. These features have expanded the use of liquid metals beyond thermal conductors and flexible electronics, opening opportunities in materials synthesis, surface engineering, and heterogeneous integration.

MOFs are highly ordered porous crystals assembled from metal nodes and organic linkers. Due to their high surface area, tunable structure, and versatile chemistry, MOFs have been widely studied for gas storage, separation, catalysis, and sensing[12–14]. Conventional methods such as hydrothermal or solvothermal synthesis can produce well-defined structures, but they often require high temperatures and long reaction times[15,16]. In recent years, attention has expanded from

[1]Department of Materials Science and Engineering, Feng Chia University, Taichung, Taiwan. [2]Department of Materials Science and Engineering, National Yang Ming Chiao Tung University, Hsinchu, Taiwan. ✉e-mail: huangcw@fcu.edu.tw

monometallic to bimetallic MOFs, where the incorporation of secondary metal centers enhances structural stability, increases the diversity of active sites, and enables improved electronic and catalytic properties. Achieving homogeneous distribution of two metal species is challenging because the differences in ionic radius, valence states, coordination numbers, and coordination geometries, as well as the variations in metal-ligand bond strength, can easily disturb balanced coordination and cause structural heterogeneity or phase segregation[17,18]. Therefore, developing a mild, controllable, and scalable synthesis strategy that enables precise bimetallic assembly is an important goal in MOF research.

Electrochemical synthesis has emerged as a promising alternative, offering ambient conditions, low energy input, and tunable growth. However, conventional electrochemical routes, including anodic dissolution and cathodic deposition, face critical limitations that hinder their broader application. Anodic dissolution is largely restricted to single-metal systems because it depends on a solid anode as the sole ion source[19,20]. For active metals such as Mg and Al, the formation of passivating layers on the anode surface severely inhibits the sustained release of ions[21]. Cathodic deposition, while capable of introducing metal ions from the electrolyte, often suffers from compositional inhomogeneity in multimetallic systems due to differences in reduction potentials and coordination kinetics, and it typically requires substrates that confine growth to electrode surfaces[22,23]. These challenges make the synthesis of bimetallic MOFs particularly difficult using conventional electrochemical strategies.

Recent studies have begun to employ liquid metals as dynamic anodes to overcome these barriers. Al-Ga and Zn-Ga systems have successfully produced single-metal MOFs, demonstrating stable and controllable ion release without passivation issues[21,24,25]. Yet, extending this concept to bimetallic MOFs with precise stoichiometric control and uniform structural integration remains unexplored, leaving a critical opportunity for advancing liquid metal interface control toward multimetallic framework synthesis.

Here, a liquid metal interface control electrochemical strategy is introduced to enable the synthesis of bimetallic ZnMg-MOF-74. By using a molten Mg-Ga alloy as the anode, $Mg^{2+}$ ions can be steadily released and co-assembled with pre-added $Zn^{2+}$ ions in solution. This dual-source approach directly overcomes the single-metal limitation of conventional anodic routes. Furthermore, the high surface tension

of the liquid metal enables electrocapillarity and Marangoni-driven transport, which allows dynamic and spatially uniform ion release[8]. As a result, a substrate-free, self-nucleating, and composition-tunable bimetallic MOF structure with good crystallinity can be achieved rapidly under ambient conditions, in contrast to hydrothermal or solvothermal synthesis that requires high temperatures and long durations. This method demonstrates the versatility of liquid metals in interfacial engineering and offers a new route for the green synthesis and structural control of multimetallic functional materials.

## Results

Conventional anodic deposition methods face inherent limitations regarding metal diversity and morphology control. Here, we introduce a liquid metal interface-controlled electrochemical strategy to construct bimetallic ZnMg-MOF-74. Unlike traditional solid metal anodes that provide a single metal source, this method uses a liquid Mg-Ga alloy as a controllable $Mg^{2+}$ donor to co-assemble with $Zn^{2+}$ in solution, demonstrating the interfacial-control potential of liquid metals in bimetallic MOF formation (Fig. 1a). In the electrolyte, $Zn(NO_3)_2·6H_2O$ dissociates to supply $Zn^{2+}$ (Eq. 1), while the linker $H_4DOBDC$ is deprotonated at the working pH to form $DOBDC^{4-}$ (Eq. 2). Upon applying an anodic bias, the Ga-Mg liquid anode continuously releases $Mg^{2+}$ via selective oxidation (Eq. 3). The simultaneously available $Zn^{2+}$, $Mg^{2+}$, and $DOBDC^{4-}$ then co-coordinate to assemble the ZnMg-MOF-74 framework (Eq. 4).

$$Zn(NO_3)_2 · 6H_2O \longrightarrow Zn^{2+} + 2NO_3^- + 6H_2O \qquad (1)$$

$$H_4DOBDC \rightleftharpoons DOBDC^{4-} + 4H^+ \qquad (2)$$

$$Mg_{(S)} \rightarrow Mg^{2+}_{(aq)} + 2e^- \qquad (3)$$

$$Mg^{2+} + Zn^{2+} + DOBDC^{4-} \longrightarrow ZnMg - MOF - 74 \qquad (4)$$

This process relies on the dynamic ion-release behavior at the liquid metal interface. Upon applying an electric field, migration of solute Mg atoms from the bulk of the Ga-Mg alloy toward the surface is

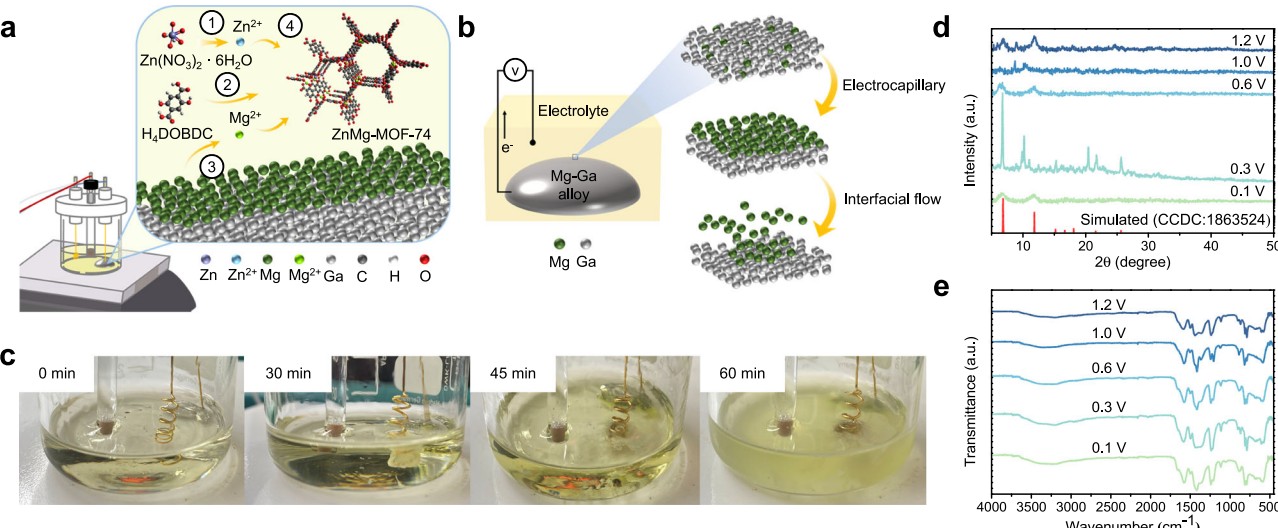

**Fig. 1 | Electrochemical synthesis and interfacial dynamics of ZnMg-MOF-74. a** Schematic illustration of the electrochemical synthesis mechanism of ZnMg-MOF-74. **b** Schematic of Mg enrichment and controlled release from the Mg-Ga liquid metal alloy under an applied electric field. **c** Schematic representation of the

suspended-state nucleation and product dispersion during MOF formation. **d** XRD patterns of ZnMg-MOF-74 synthesized at different applied voltages. **e** FTIR spectra of ZnMg-MOF-74 synthesized under varying voltages.

accelerated by electromigration, establishing an internal concentration gradient within the liquid metal[9,24]. Together with the intrinsically high surface tension of Ga, this gradient induces electrocapillarity, which modulates the interfacial energy and promotes Mg enrichment at the anode surface[8]. The resulting perturbation of surface tension further generates a surface-tension gradient that drives Marangoni convection, continuously carrying $Mg^{2+}$ from the alloy surface into the surrounding electrolyte (Fig. 1b)[26]. Benefiting from the unique physicochemical properties of gallium, this mechanism effectively prevents anodic passivation that typically occurs with highly reactive metals such as Mg. The Ga-based liquid metal maintains a conductive and self-healing surface that supports continuous $Mg^{2+}$ release and transport, enabling long-term reaction stability and uniform ion supply during electrochemical MOF formation.

Supplementary Fig. 1 shows the cyclic voltammetry (CV) results of pure Ga and Ga-Mg alloy electrodes. The pure Ga electrode exhibits no observable redox peaks, confirming its electrochemical inertness. In contrast, the Mg-containing alloy displays an anodic peak near + 1.4 V, indicating Mg oxidation and supporting the proposed preferential ion release mechanism. Furthermore, Supplementary Fig. 2 presents the time-current profiles recorded during synthesis under various applied voltages. The steady current responses reflect stable ion release behavior, indirectly supporting the Marangoni-driven redistribution and sustained availability of Mg ions at the reactive interface[9].

In line with this mechanism, Supplementary Table 1 summarizes representative bimetallic MOF syntheses reported in recent years. Most prior routes rely on solvothermal or ion-exchange methods conducted at elevated temperatures for extended durations, and they are often constrained by substrate dependence or compositional inhomogeneity[17,18]. By contrast, the present liquid metal interface strategy achieves rapid assembly under ambient conditions without substrates, while allowing flexible and tunable control of metal stoichiometry. These distinctions highlight the capability of liquid metals to overcome long-standing challenges in multimetallic MOF synthesis.

Figure 1c shows that time-resolved observations reveal that the product initially forms on the counter electrode and gradually detaches into the solution. After 60 min, the solution becomes uniformly turbid, indicating that the strategy allows substrate-independent nucleation and suspended-state MOF formation, beyond the surface confinement of conventional electrochemical synthesis. To identify representative synthesis conditions, the effects of alloy ratio, ligand concentration, and reaction time were evaluated (Supplementary Figs. 3–5). At 1.0 wt% Mg, MOF-74 peaks appeared clearly. In contrast, using 1.5 wt% Mg required higher voltage to form the framework, suggesting that higher Mg levels need stronger kinetics to drive crystal growth, which limits its suitability for systematic studies (Supplementary Fig. 3). Among tested ligand concentrations, only 0.03 M yielded well-defined MOF-74 patterns (Supplementary Fig. 4). For reaction time, 60 min was sufficient to form stable structures, with no major improvement at longer durations (Supplementary Fig. 5). Based on these results, the condition of 1.0 wt% Mg, 0.03 M ligand, and 60 min was selected for voltage-dependent synthesis.

Figure 1d displays the characteristic diffraction peaks of MOF-74 under all tested conditions, confirming crystalline phase formation driven by the applied electric field. Notably, at 0.3 V, the main peak at $2\theta = 6.8°$ appears with the highest sharpness, and a secondary peak shifts from 11.8° to 10.1°, likely reflecting differences in nucleation orientation or bimetallic composition, indicating improved crystallinity and structural integrity under this condition[27]. Fourier transform infrared spectroscopy (FTIR) spectra (Fig. 1e) further confirm successful metal-ligand coordination. The asymmetric and symmetric stretching of $-COO^-$ appear at 1595 and 1418 $cm^{-1}$, respectively, along with distinct vibrations from C-O and C-H in the aromatic ring. Additional bands at 583 $cm^{-1}$ and 605 $cm^{-1}$ correspond to Zn-O and Mg-O,

confirming that both metal ions participate in the framework and form a bimetallic MOF structure[28].

The influence of applied voltage on the morphology and composition of ZnMg-MOF-74 was systematically investigated to understand how electrochemical parameters affect interfacial control (Fig. 2a–g). By adjusting the electric field strength, both the metal ion release rate and nucleation kinetics could be modulated, thereby influencing framework assembly. Field emission scanning electron microscopy (FE-SEM) images show that under a low voltage of 0.1 V (Fig. 2a), the product mainly consists of loose and disordered aggregates, suggesting insufficient nucleation for crystal growth. At 0.3 V (Fig. 2b), a well-aligned nanosheet morphology appears across the surface. This indicates that the reaction rate under this condition was moderate and stable, allowing for proper lattice organization and directional growth. It was identified as the optimal morphology in this study. When the voltage increased to 0.6 V and 1.0 V (Fig. 2c, d), the morphology changed into irregular clusters, likely due to excessive nucleation that caused strong competition between nuclei and limited crystal growth. At 1.2 V (Fig. 2e), some sheet-like features were still visible, but the overall product became densely packed and disordered, suggesting that the reaction dynamics no longer supported ordered crystallization. Energy dispersive X-ray spectroscopy (EDS) analysis revealed how the Zn:Mg ratio varied with voltage (Fig. 2f). A nearly 1:1 ratio was found at 0.3 V, matching the formation of a uniform nanosheet structure[29]. In contrast, samples at 0.1 V and 0.6-1.0 V showed excess Mg, with Zn:Mg dropping to about 1:2 or lower. This imbalance may have led to amorphous $Mg(OH)_2$ formation, which can hinder ordered MOF growth. Although the Zn:Mg ratio returned to ~ 1:1 at 1.2 V, the fast reaction rate and early aggregation of nuclei still prevented the formation of stable sheets. These results suggest that both composition and reaction kinetics must be well balanced to achieve high crystallinity. Elemental mapping was carried out for the sample at 0.3 V to evaluate nanosheet homogeneity (Fig. 2g). Zn, Mg, and O were uniformly distributed across the structure, confirming the cooperative integration of both metal ions at the microscale. Notably, no Ga signal was detected, indicating that Ga acted only in interfacial regulation within the liquid alloy without being oxidized or embedded into the framework. This behavior matches Ga's high standard reduction potential and inert chemical nature. Ga plays two important roles: maintaining the electrical conductivity and anodic stability of the system, and generating electrocapillarity and Marangoni flow through its high surface tension. These effects enable stable and uniform metal ion release. This kind of "selective participation with functional stability" is a key advantage of liquid metal systems and underlies their potential in interface-controlled material synthesis.

A series of characterizations, including X-ray photoelectron spectroscopy (XPS), Brunauer–Emmett–Teller (BET), and thermogravimetric analysis (TGA), were performed to evaluate the chemical composition, porosity, and thermal properties of the synthesized ZnMg-MOF-74 (Fig. 3a–f). The XPS survey spectrum confirmed the presence of Mg, Zn, O, and C, with no detectable impurities (Fig. 3a), indicating a clean synthesis process. High-resolution spectra further clarified the oxidation states of the metal species. The Zn $2p_{3/2}$ and $2p_{1/2}$ peaks appeared at 1021.4 eV and 1044.7 eV, respectively (Fig. 3b), consistent with $Zn^{2+}$. The Mg 1 $s$ peak at 1303.8 eV, along with Auger features at 305.2 eV and 350.8 eV (Fig. 3c), confirmed the presence of $Mg^{2+}$. The O 1 $s$ spectrum (Fig. 3d) showed three components: metal-oxygen bonds (530.2 eV), C-O groups (531.8 eV), and adsorbed water or hydroxyls (533.8 eV)[28,30]. These results support the formation of coordinated metal-ligand bonds and indicate some surface chemical diversity.

The porosity of ZnMg-MOF-74 was analyzed by nitrogen adsorption-desorption isotherms (Fig. 3e). The $N_2$ adsorption-desorption curve exhibits a Type IV profile with an H3 hysteresis loop, characteristic of mesoporous materials and indicative of slit-

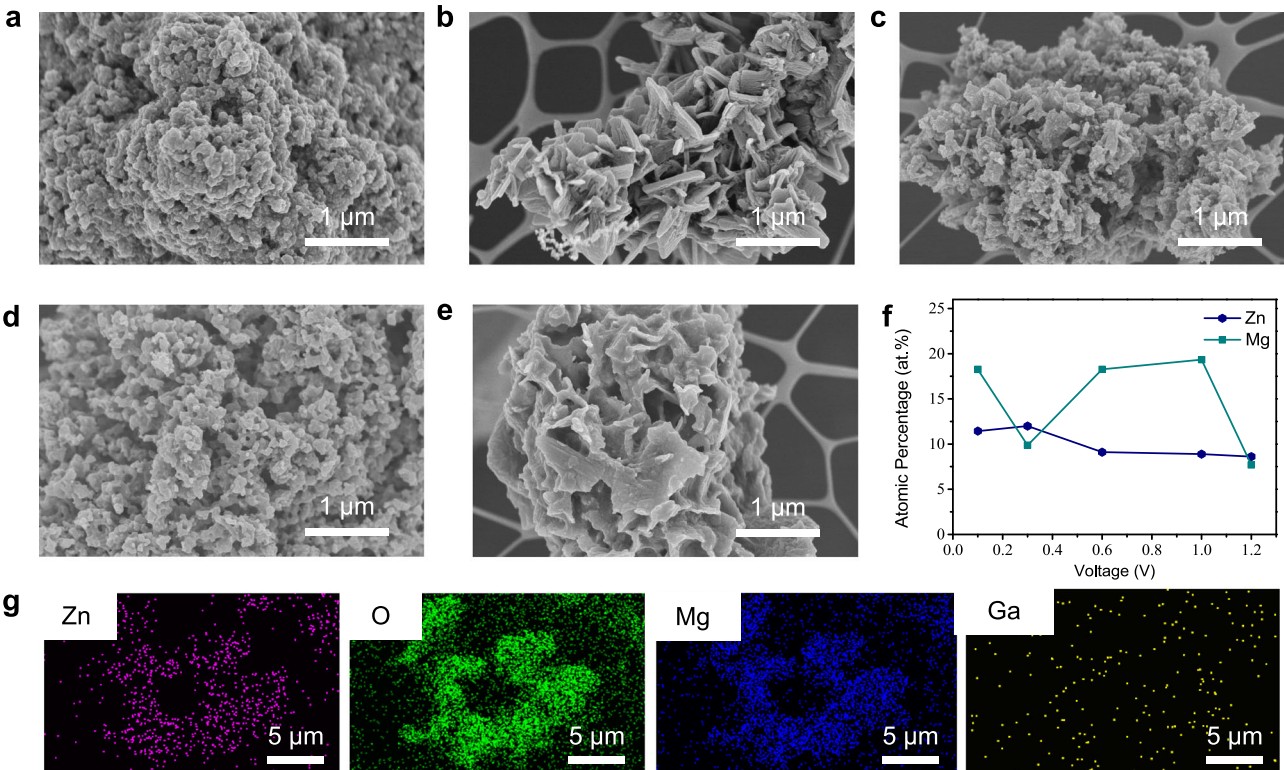

**Fig. 2 | Voltage-dependent morphology and composition of ZnMg-MOF-74. a–e** SEM images of ZnMg-MOF-74 synthesized at different voltages: (**a**) 0.1 V; (**b**) 0.3 V; (**c**) 0.6 V; (**d**) 1.0 V; **e** 1.2 V. **f** Elemental Zn:Mg ratio of products synthesized under various voltages. **g** Elemental mapping of Zn, Mg, and O in the sample synthesized at 0.3 V.

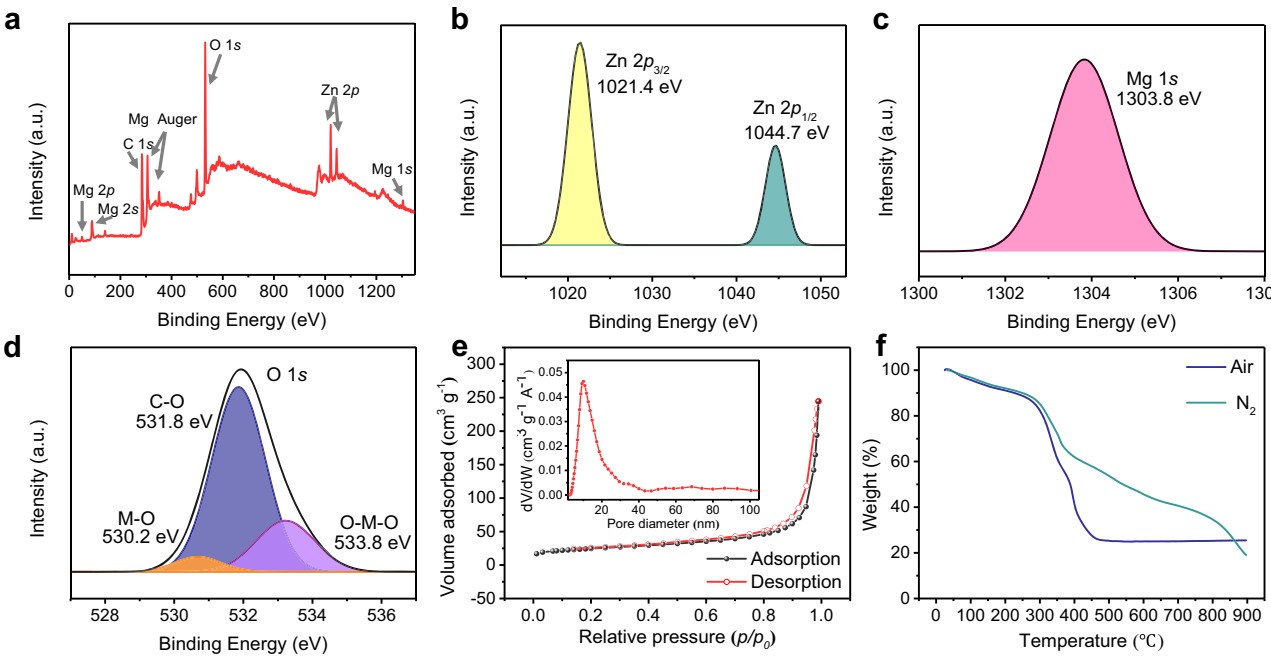

**Fig. 3 | Composition, porosity, and thermal stability of ZnMg-MOF-74. a** XPS survey spectrum. **b–d** High-resolution XPS spectra of (**b**), Zn 2*p*; (**c**), Mg 1*s*; (**d**, O 1 s. **e** N$_2$ adsorption-desorption isotherm measured at 77 K. Inset: corresponding pore size distribution curve calculated using the BJH method. **f** TGA under N$_2$ and air atmospheres.

shaped mesopores formed by interwoven sheet-like aggregates. This feature corresponds well with the sheet-interlaced morphology observed in SEM (Fig. 2b), confirming the consistency between pore structure and particle arrangement. The sample exhibited a BET surface area of 210.92 m² g⁻¹, with pore sizes ranging from 1.4 to 30 nm and an average pore diameter of 12.06 nm, reflecting a stable mesoporous framework favorable for mass transport. These values are

consistent with the one-dimensional channels of MOF-74 and suggest favorable mass transport and carrier mobility in potential optoelectronic applications. Thermal stability was examined using thermogravimetric analysis (Fig. 3f). Under a nitrogen atmosphere, a slight weight loss occurred between 100–200 °C due to the removal of adsorbed water and residual solvents. A primary decomposition followed between 300–600 °C, attributed to the breakdown of the

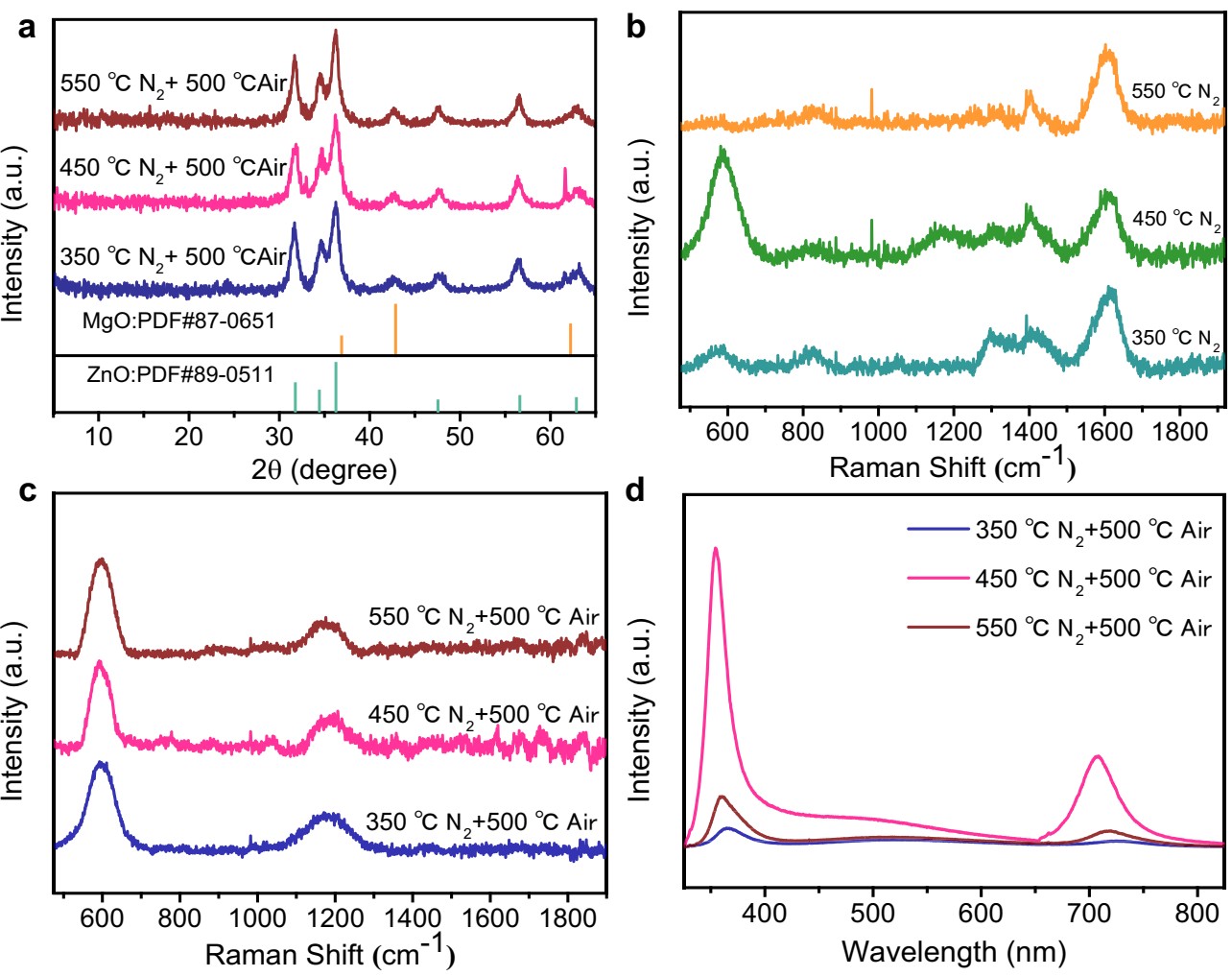

**Fig. 4 | Structural transformation and optical properties of MOF-derived ZnMgO. a** XRD pattern after two-step annealing. **b** Raman spectra after the first-step annealing under $N_2$. **c** Raman spectra after the second-step annealing in air. **d** Photoluminescence (PL) spectra of ZnMgO after two-step annealing.

organic linkers, after which the sample stabilized above 700 °C, suggesting the formation of carbonaceous residues[28]. In contrast, samples heated in air decomposed rapidly between 250–400 °C, leading to complete structural collapse. This contrast highlights the necessity of a two-step annealing process: the first stage under an inert atmosphere generates a relatively stable metal-carbon intermediate that preserves the framework morphology, while the subsequent stage in air removes the residual carbon and yields crystalline oxides[31]. The resulting product, denoted as ZnMgO, refers to the mixed ZnO-MgO oxide derived from the ZnMg-MOF-74 precursor, rather than a single-phase compound. Therefore, the atmosphere not only determines the thermal decomposition pathway but also governs the crystallinity and morphology of the derived ZnMgO, explaining why two-step annealing provides better structural retention than one-step air annealing. Supplementary Fig. 6 further supports this, showing that air-treated samples adopted an amorphous morphology in TEM, highlighting the critical role of atmosphere in determining the structural transformation outcome.

Following the structural validation of ZnMg-MOF-74, we investigated its phase transformation behavior during high-temperature annealing using XRD and Raman spectroscopy. X-ray diffraction patterns (Fig. 4a) revealed distinct peaks corresponding to ZnO (JCPDS No. 89-0511), including the (100), (002), (102), (110), and (103) planes at 2θ = 31.7°, 34.6°, 47.6°, 56.6°, and 63.1°, respectively. A peak at 42.8° was also observed, matching the (200) plane of MgO (JCPDS No. 87-

0651), confirming the complete conversion of the precursor MOF into a ZnO-MgO bimetallic oxide[32]. No residual signals from the MOF or impurities were detected, indicating complete decomposition of the organic framework.

To gain insights into the nucleation process and ligand breakdown, Raman spectra were collected after nitrogen pretreatment (Fig. 4b). At 350 °C, a weak $A_1$(LO) mode of Zn-O appeared at 580.48 cm$^{-1}$, suggesting the onset of ZnO crystallization. At the same time, Zn ions remained partially embedded in the organic ligands[33]. A band at 828.19 cm$^{-1}$, attributed to out-of-plane bending of aromatic rings, also indicated incomplete ligand decomposition[34]. When the temperature increased to 450 °C, the $A_1$(LO) mode blue-shifted to 590 cm$^{-1}$ and its intensity rose significantly, reflecting improved lattice order. This behavior is consistent with classical nucleation theory:

$$\Delta G_T = \Delta G_V + \Delta G_S \tag{5}$$

The system likely surpassed the critical energy barrier (ΔG*) at this stage, enabling stable crystal nucleation and lattice rearrangement. Meanwhile, new peaks emerged at 1164 cm$^{-1}$ and 1295.8 cm$^{-1}$, corresponding to C-O and C-N vibrations, which were associated with thermolysis intermediates and further ligand decomposition[34]. At 550 °C, however, the $A_1$(LO) peak vanished, suggesting that while nucleation continued, high reaction rates and rapid ion diffusion suppressed crystal growth and reduced lattice order. This can be

explained using the critical radius model:

$$r^* = \frac{2r}{\Delta G_V} \tag{6}$$

As $\Delta G_V$ increases with temperature, $r^*$ decreases, leading to more nucleation sites but smaller crystals, which results in lower Raman intensity and diminished crystallinity. After post-annealing in air at 500 °C, Raman spectra (Fig. 4c) showed that both D and G bands nearly disappeared, confirming complete oxidation and removal of carbon residues from the original MOF ligands. This further supports the transformation into an inorganic oxide phase. These observations clarify the sequential nature of the two-step annealing strategy. In the first stage under $N_2$, cleavage and carbonization of the organic linkers yield a Zn/Mg-C intermediate that preserves the overall morphology and prevents abrupt collapse. In the second stage under air, the carbon matrix is oxidized and removed, enabling Zn and Mg to crystallize separately into ZnO and MgO domains. A schematic of this conversion pathway is provided in Supplementary Fig. 7 to aid interpretation of the atmosphere-controlled transformation. Photoluminescence (PL) measurements (Fig. 4d) showed a near-band-edge (NBE) emission in the UV region, consistent with intrinsic bandgap transitions of ZnO. When the pretreatment temperature increased from 350 °C to 450 °C, the NBE peak blue-shifted and became stronger. This was attributed to the Burstein-Moss effect caused by oxygen vacancies, which led to Fermi level shifting and apparent bandgap widening. Improved crystallinity at this stage also enhanced PL intensity by reducing non-radiative recombination. In contrast, at 550 °C, the NBE peak red-shifted and weakened, indicating that vacancy recombination or defect compensation lowered free carrier density and light emission efficiency.

In the visible region, all samples exhibited clear defect-related emission, mainly caused by deep-level oxygen vacancies and lattice imperfections. The 450 °C sample showed the strongest emission, implying that this condition favored stable oxygen vacancy formation. At 550 °C, these features weakened, likely due to defect healing or the formation of disordered structures.

High-resolution transmission electron microscopy (HR-TEM) and Fast Fourier Transform (FFT) analyses were carried out to visualize grain evolution in ZnMgO under different pretreatment conditions and correlate the microstructure with optoelectronic performance (Fig. 5). The TEM images reveal that after two-step annealing, ZnO and MgO crystallize as distinct domains separated by amorphous regions, rather than forming a uniform mixed phase. This result is in line with the two-step annealing process shown in Supplementary Fig. 7, clarifying why crystalline ZnO and MgO domains are interspersed with amorphous regions. At low magnification, EDS mapping (Supplementary Fig. 8) confirms that Zn and Mg remain homogeneously distributed at the particle scale, indicating macroscopic compositional uniformity, while high-resolution TEM highlights the presence of localized crystalline grains. All samples underwent final air annealing at 500 °C, following thermal pretreatment at 350 °C, 450 °C, or 550 °C in nitrogen. After 350 °C pretreatment (Fig. 5a), TEM images showed evenly distributed grains with clear lattice fringes, indicating successful crystallization into oxide phases. FFT analysis identified the ZnO $(1\bar{1}0)$ plane with a spacing of 0.271 nm along the [001] zone axis, and the MgO (020) plane with 0.249 nm spacing along $[10\bar{1}]$. The average grain area was 7.17 nm² (Supplementary Table 2), suggesting that nucleation had begun but grain growth remained limited. As a result, the optoelectronic performance was moderate, with a responsivity (R) of $7.64 \times 10^{-4}$ A/W, external quantum efficiency (EQE) of $2.60 \times 10^{-5}$, and detectivity (D*) of $2.60 \times 10^{-5}$ Jones (Supplementary Table 3), implying that incomplete crystallinity restricted carrier transport and light response.

In comparison, the 450 °C pretreatment (Fig. 5b) led to significant improvements in grain size and crystallinity. HR-TEM images showed ZnO (011) planes with a spacing of 0.244 nm (zone axis $[11\bar{1}]$) and MgO (002) planes with 0.254 nm spacing (zone axis $[\bar{1}\bar{1}0]$). The average grain area increased to 27.88 nm² (Supplementary Table 2), the largest among the three conditions. This enhancement corresponded to the best optoelectronic results, with R reaching 1.48 A/W, EQE of 5.04%, and D* of $4.08 \times 10^9$ Jones (Supplementary Table 3), suggesting that this condition supported stable nucleation and grain growth, leading to better charge transport and photoconversion efficiency. By contrast, the 550 °C pretreatment sample (Fig. 5c) still exhibited crystalline ZnO and MgO phases. HR-TEM images revealed ZnO (011) planes with a spacing of 0.254 nm along the $[1\bar{1}1]$ zone axis and MgO $(11\bar{1})$ planes with 0.243 nm spacing along the [011] zone axis. However, the average grain area sharply decreased to 3.66 nm² (Supplementary Table 2), indicating suppressed grain growth. This behavior likely resulted from rapid thermolysis in nitrogen, which produced highly dispersed oxide precursors and limited local aggregation of metal atoms needed for stable nucleation. Although crystal formation occurred during the final air annealing, the resulting grains were small and fragmented, reducing charge transport pathways. Accordingly, the optoelectronic performance declined, with R, EQE, and D* falling to $5.26 \times 10^{-6}$ A/W, $1.79 \times 10^{-5}$ %, and $1.03 \times 10^8$ Jones (Supplementary Table 3), respectively—much lower than the values achieved under 450 °C pretreatment.

The UV photodetection performance of ZnMgO was evaluated under different pretreatment conditions to assess its potential for high-speed optoelectronic applications (Fig. 6a–c).

Among the three samples, ZnMg-450$N_2$-500Air exhibited the lowest dark current, indicating a lower intrinsic carrier concentration. This behavior is attributed to a reduced density of structural defects and a moderate level of oxygen vacancies. In contrast, the ZnMg-350$N_2$-500Air and ZnMg-550$N_2$-500Air samples showed significantly higher dark currents, likely due to increased grain boundary density or excessive shallow-level defects that compromised the device's static stability.

Under UV illumination, ZnMg-450$N_2$-500Air produced a much higher photocurrent than the other two, reflecting superior photocarrier generation and transport. This enhancement aligns with the TEM results, which showed that this sample had the largest average grain size (27.88 nm²) and a more continuous grain network—both favorable for promoting carrier mobility and suppressing non-radiative recombination. Consistent with the PL results in Fig. 4d, the differences in photoresponse among these samples can also be attributed to the evolution of oxygen-related defects. At 450 °C, the blue-shifted and stronger NBE indicates a higher free-carrier density, which facilitates charge transport and explains the enhanced responsivity and slower decay. In contrast, at 550 °C, the red-shifted and weakened NBE reflects defect compensation or partial healing of oxygen vacancies, reducing carrier density and accelerating recombination; the concurrent weakening of visible defect emission is also consistent with fewer conductive defect channels. Therefore, the photoresponse trends are governed not only by grain size and microstructural continuity but also by the concentration and activity of oxygen-related defects, in agreement with the PL analysis. In contrast, ZnMg-550$N_2$-500Air, with its fragmented and fine grains (~3.66 nm²), exhibited poor charge transport and a much lower photocurrent.

To assess switching stability and temporal response, periodic on/off measurements were performed under a −4 V bias and 365 nm UV light (Fig. 6d–f). The ZnMg-450$N_2$-500Air device showed stable and repeatable current switching in each cycle, confirming excellent photoresponse stability and reliable operation. Further time-resolved analysis (Fig. 6g–i) revealed fast rise and decay times of 0.32 s and 0.64 s, respectively—significantly faster than the 1.21 s and 1.22 s observed in the 350 °C and 550 °C samples. These results demonstrate

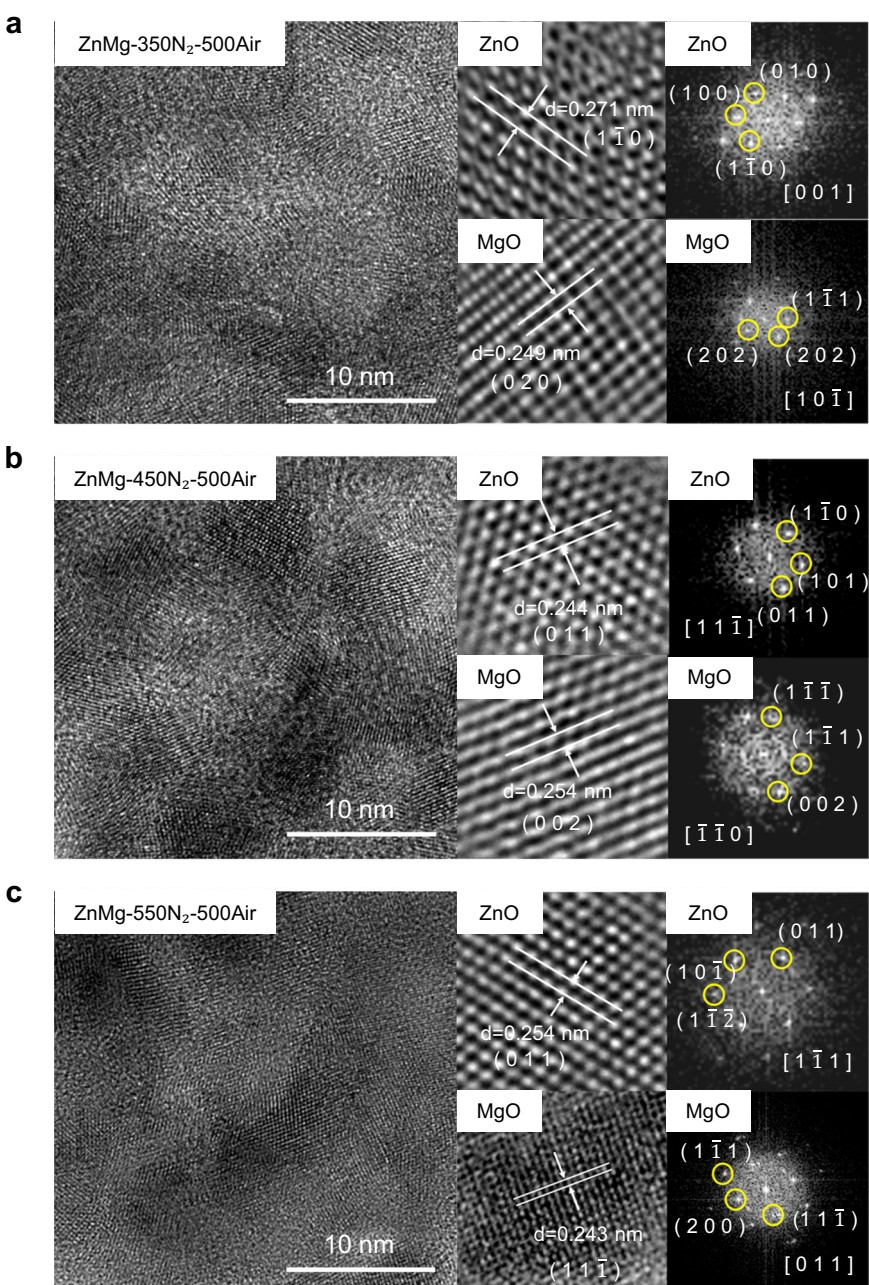

**Fig. 5 | Microstructure evolution of ZnMgO derived from ZnMg-MOF-74 under different annealing conditions. a** HR-TEM image and corresponding FFT pattern of ZnMgO pretreated in $N_2$ at 350 °C followed by air annealing at 500 °C. The sample shows clear lattice fringes corresponding to ZnO (1$\bar{1}$0) planes with an interplanar spacing of 0.271 nm (zone axis = [0 0 1]) and MgO (0 2 0) planes with 0.249 nm spacing (zone axis = [1 0 $\bar{1}$]). **b** HR-TEM image and FFT pattern of ZnMgO pretreated in $N_2$ at 450 °C and annealed in air at 500 °C. Well-defined lattice fringes are identified as ZnO (0 1 1) planes with an interplanar spacing of 0.244 nm (zone axis = 1 1 $\bar{1}$]) and MgO (0 0 2) planes with 0.254 nm spacing (zone axis = [$\bar{1}\bar{1}$0]). **c** HR-TEM image and FFT pattern of ZnMgO pretreated in $N_2$ at 550 °C and annealed in air at 500 °C. Distinct lattice fringes are observed for ZnO (0 1 1) planes with an interplanar spacing of 0.254 nm (zone axis = [1$\bar{1}$1]) and MgO (11$\bar{1}$) planes with 0.243 nm spacing (zone axis = [0 1 1]).

that ZnMg-450N$_2$-500Air features fast and reversible carrier dynamics under UV excitation, positioning it as a strong candidate for high-speed ultraviolet photodetectors.

Beyond the Zn-Mg system demonstrated here, the concept of liquid metal interface control is generalizable to other alloys and strategies. In line with the emerging view of atomic intelligence in liquid metals, the dynamic and reconfigurable active sites at Ga-based interfaces, together with surface-tension-driven flow and electro-capillarity, enable selective ion release, uniform atomic dispersion, and stable operation that are difficult to realize with solid electrodes[35]. In our platform, alloying provides a practical handle to program which species is oxidized at the interface: metals with lower standard potentials than Ga can be selectively released as cations under bias. Accordingly, Ga-based eutectics incorporating Al, Zn, Mn, or Cd can serve as alternative ion sources with tunable electrochemical driving forces, while dynamically reconfigurable, high-surface-energy liquid metals may broaden the accessible palette of transition-metal constituents. Rational selection of metal pairs should also consider ionic radius and coordination compatibility in the target framework. In MOF-74, $Zn^{2+}$ (0.74 Å) and $Mg^{2+}$ (0.72 Å) have closely matched

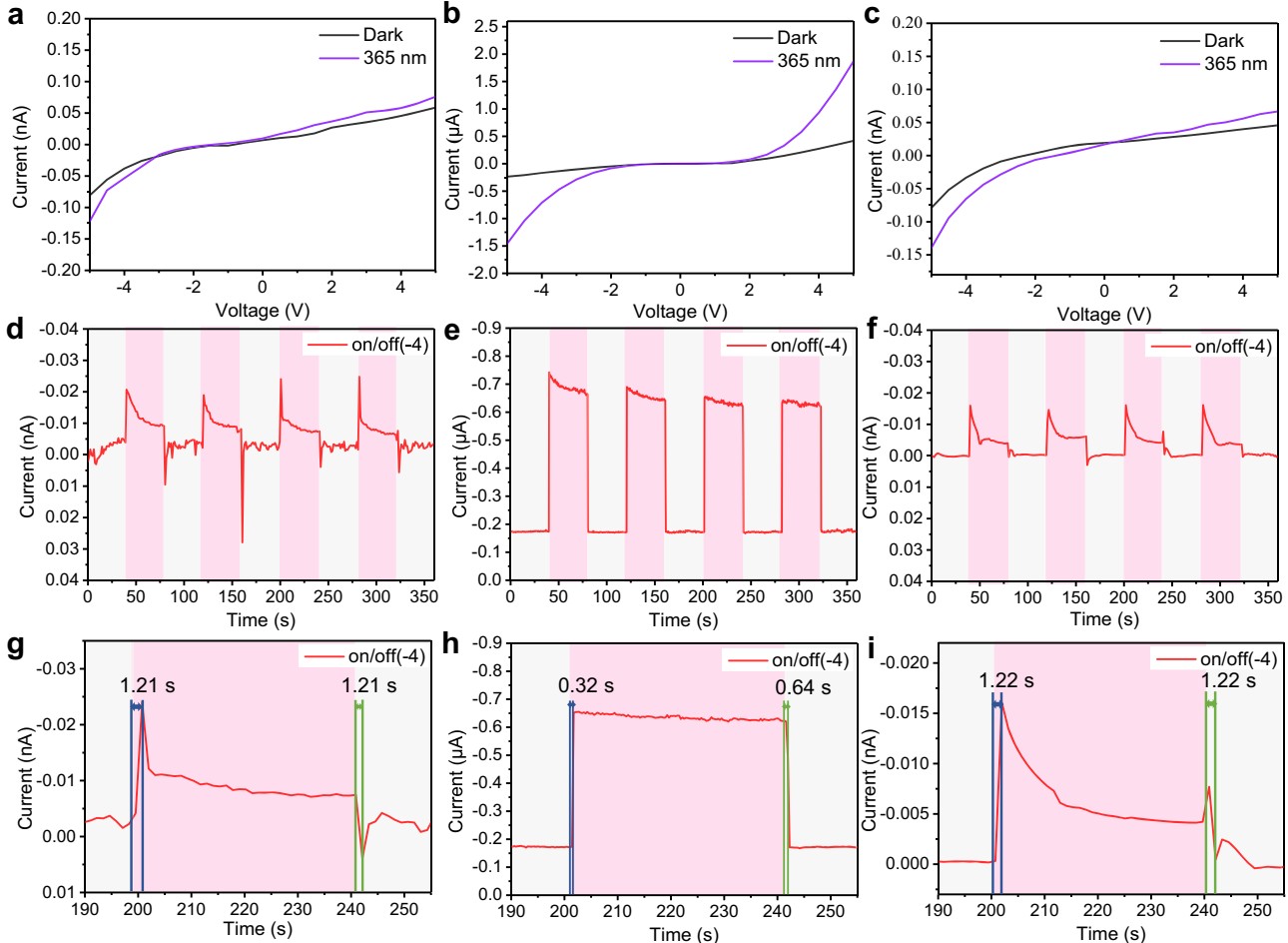

**Fig. 6 | Optoelectronic performance of ZnMgO photodetectors under different pretreatment temperatures. a**–**c** I-V characteristics of ZnMgO photodetectors derived from ZnMg-MOF-74 precursors pretreated in $N_2$ at 350 °C, 450 °C, and 550 °C, measured under dark and 365 nm UV illumination. ZnMg-450 $N_2$-500 Air shows the lowest dark current and the highest photocurrent. **d**–**f** Periodic on/off photoresponse cycles of ZnMg-350 $N_2$-500 Air, ZnMg-450 $N_2$-500 Air, and ZnMg-550 $N_2$-500 Air measured under a − 4 V bias and 365 nm UV illumination. Each illumination cycle was switched every 40 s. The gray regions represent the dark state, and the pink regions represent the 365 nm UV illumination state. ZnMg-450 $N_2$-500 Air exhibits relatively stable and reproducible photoresponse behavior. **g**–**i** Time-resolved photoresponse profiles of ZnMg-350 $N_2$-500 Air, ZnMg-450 $N_2$-500 Air, and ZnMg-550 $N_2$-500 Air. ZnMg-450 $N_2$-500 Air exhibits the fastest carrier dynamics with a rise time of 0.32 s and a decay time of 0.64 s.

radii, which supports the successful formation of the bimetallic framework reported here. Similar size and coordination chemistry likewise make Zn-Co and Zn-Ni combinations feasible candidates for bimetallic MOFs. We anticipate that these alloy choices and control levers will extend interface-directed electrosynthesis beyond Zn/Mg, expanding both the structural tunability and functional scope of bimetallic and multimetallic MOFs and their derived oxides, and motivating future in situ studies to resolve interfacial mechanisms.

## Discussion

This study presents a demonstration of a liquid metal interface–controlled electrochemical strategy for synthesizing bimetallic MOFs, overcoming the conventional limitations of anodic deposition that rely on single-metal sources and solid substrates. By precisely regulating metal ion release and nucleation kinetics, a ZnMg-MOF-74 framework was successfully constructed and further converted into a ZnMgO composite oxide with high photoresponse performance. The results reveal that grain size, lattice ordering, and oxygen vacancy distribution are key microstructural factors governing optoelectronic behavior.

This synthesis platform offers advantages such as tunable composition, fast reaction kinetics, and substrate-free processing, showing strong potential for interface engineering and material generality. Future developments may extend this strategy to various multimetallic MOFs and their derivatives for applications in gas sensing, low-power optoelectronics, and energy-related catalysis, suggesting broad potential for liquid metal chemistry in advanced material systems.

## Methods

### Preparation of liquid metal alloy

A Mg-Ga alloy was prepared with a weight ratio of Ga:Mg = 99:1. Metallic gallium (99 wt%) and magnesium (1 wt%) were placed in an alumina boat and heated under argon atmosphere to 750 °C at a rate of 10 °C/min. The mixture was held at this temperature for 2 h to ensure homogeneous alloying. The resulting liquid alloy was transferred into a sealed glass vial and stored at 50 °C for subsequent electrochemical synthesis.

### Electrochemical synthesis of ZnMg-MOF-74

Electrochemical synthesis was performed in a three-electrode system using a saturated calomel electrode (SCE) as the reference electrode and

two gold electrodes as the working and counter electrodes. The working electrode was brought into contact with the pre-prepared Mg-Ga liquid alloy. The electrolyte consisted of 0.03 M 2,5-dihydroxyterephthalic acid ($H_4DOBDC$), 0.1 M tetrabutylammonium hexafluorophosphate ($NBu_4PF_6$), and 0.1 M zinc nitrate hexahydrate ($Zn(NO_3)_2 \cdot 6H_2O$), dissolved in 10 mL of N,N-dimethylformamide (DMF).

The synthesis was carried out under a constant potential at 50 °C for 60 min using a potentiostat (BioLogic, model SP-300). After the reaction, the product was collected by centrifugation at $5500 \times g$ and washed three times each with DMF and absolute ethanol. The resulting powder was dried under vacuum at 60°C overnight to yield ZnMg-MOF-74.

### Preparation of MOF-Derived Oxides
The obtained ZnMg-MOF-74 powder was converted into metal oxides through a two-step annealing process. In the first step, the sample was placed in an alumina boat and annealed under nitrogen at 350 °C, 450 °C, or 550 °C (3 °C/min, 2 h holding). This step removed the organic ligands and partially retained carbon species. In the second step, the sample was further annealed in air at 500 °C (3 °C/min, 2 h) to complete the oxidation process.

### Fabrication of electrical devices
Low-resistivity n-type silicon wafers were used as substrates. After RCA cleaning and ultrasonic treatment, a ~ 200 nm $SiO_2$ dielectric layer was deposited by plasma-enhanced chemical vapor deposition (PECVD). Interdigitated electrode patterns were defined by photolithography, followed by sequential deposition of a 20 nm Ti adhesion layer and a 100 nm Au electrode layer using magnetron sputtering. After lift-off, the devices were used for subsequent electrical and optoelectronic characterizations.

### Characterization of structure
The morphology of the samples was examined using field emission scanning electron microscopy (FE-SEM, HITACHI, model PISC019) and field-emission transmission electron microscopy (FE-TEM, JEOL, model JEM-F200). Crystalline phases were identified by X-ray diffraction (XRD, Bruker, model D8 Discover). Surface chemical bonding states and elemental compositions were analyzed via X-ray photoelectron spectroscopy (XPS, ULVAC PHI, Versa Probe 4). Molecular vibrations and chemical functionalities were further characterized using Fourier-transform infrared spectroscopy (FTIR, PERKIN ELMER, model Frontier) and confocal micro-Raman spectroscopy (Jinghong Technology Co., Ltd., UniDRON Laser Spectroscopy). The specific surface area and pore size distribution were measured by nitrogen adsorption-desorption isotherms using a BET analyzer (Micromeritics, ASAP2020). Thermal stability was evaluated by thermogravimetric analysis (TGA, TA Instruments, model TGA2950).

### Electrical and optical characterizations
The current-voltage (I–V) characteristics and conductivity of the samples were measured using a semiconductor parameter analyzer (KEITHLEY, model 2634B) in conjunction with a microprobe system equipped for low-current (100 fA) and variable-temperature measurements (ADVANCED). For optoelectronic measurements, a UV point light source (Brightek, BK Spotcuer 100 UV LED) was employed to evaluate the photodetection response under UV illumination.

## Data availability
All data supporting the findings of this study are provided in the main text and Supplementary Information. Additional data are available from the corresponding authors upon request. Source data are provided in this paper.

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

## Acknowledgements

The authors acknowledge the support from the National Science and Technology Council (NSTC) in Taiwan through the following grants: NSTC 112-2221-E-035-021-MY3 (C.-W. H.) and 114-2628-E-035 -001 -MY4 (C.-W. H.).

## Author contributions

J.-C.L. and C.-W.H. designed the study. J.-C. L. conducted the experiments. J.-C.L., C.-T.W., and C.-W.H. analyzed the data and revised the manuscript. C.-H.W. and W.-W.W. performed TEM characterization and analysis. Y.-C.C. and W.-W.W. provided experimental resources. J.-C.L. and C.-W.H. wrote the manuscript. All authors discussed the results and edited the manuscript.

## Competing interests

The authors declare no competing interests.
