## [Peer Review File · Nature Communications]

Synthesis of Bimetallic MOFs via Interface Control Using Gallium-Based Liquid Metal

Corresponding Author: Professor CHUNWEI HUANG

Version 0:

Reviewer comments:

Reviewer #1

(Remarks to the Author)

The authors presented a liquid metal based electrochemical process for synthesizing bimetallic ZnMg-13 MOF-74.

They applied liquid gallium as the anode Mg^{2+} released 15 and co-assembled with Zn^{2+} to form bimetallic framework. At 0.3 V they obtained with nanosheets with crystallinity. After annealing, the derived ZnMgO used for photodetection.

It is an interesting paper

I only have minor comments:

1- Figure 3 N_2 adsorption-desorption isotherm is not well presented. Also the inset is not readable. I suggest that the authors repeat this measurements

2-More discussion on TGA is required. The current discussion may not be quite accurate.

3- The photodetection responses reasons presented are not the most convincing for Fig 6 a to c differences may not be the once included in the paper. Maybe other suggestions are required for the differences

4- A final discussion on whether other alloys and strategies can help to expand the concept (use ref Science 389, 372, 2024)

5- Please expand the results and discussion parts

Reviewer #2

(Remarks to the Author)

The topic of this manuscript, liquid metals, MOFs, and electrochemical synthesis, is of significant interest to the materials science and engineering community.

For publication in Nature Communications, the novelty aspect of the manuscript needs to be more convincing. Since bimetallic MOFs have been synthesized previously, the authors should explicitly contrast their approach with existing methods. How does their method provide advantages in structural control, stability, or functionality compared with other reported strategies?

The use of surface-tension-driven flow to regulate Mg ion release is elegant. However, if Mg ions are released at the liquid metal–solution interface, one would expect a concentration gradient extending into the solution. Did the authors probe this experimentally, e.g., through spatially resolved spectroscopy or compositional mapping? Were changes in MgZnO composition or morphology observed as a function of distance from the liquid metal surface? Please make this point clearer in the manuscript.

The use of equations 1 and 2 to explain nucleation and crystal growth does not add new understanding in electrosynthesis of the bimetallic MOF crystal. Are MgO and ZnO synthesized separately or co-dependently? What is the critical nucleus size of this system?

Figure 5 presents TEM data showing distinct ZnO and MgO crystalline phases but does not provide information on their spatial distribution. How uniform is the MgZnO structure overall? This is also related to the above question. Are the MgO and

ZnO domains intimately intermixed, or do they exist as heterogeneous, segregated grains? This distinction is important, as microstructural uniformity would strongly influence potential applications (e.g., in catalysis, sensing, or electronics). High-resolution compositional mapping could help address this point. There is a missed opportunity to provide in-depth understanding of the bimetallic MOF microstructure and nucleation mechanism in contrast to single metal MOF nucleation and crystal growth mechanisms.

Version 1:

Reviewer comments:

Reviewer #1

(Remarks to the Author)

All OK

Reviewer #2

(Remarks to the Author)

I am satisfied with the response and corrections made by the authors in the revised manuscript.

Response to Reviewer 1:

The authors presented a liquid metal based electrochemical process for synthesizing bimetallic ZnMg-MOF-74.

They applied liquid gallium as the anode Mg^{2+} released 15 and co-assembled with Zn^{2+} to form bimetallic framework. At 0.3 V they obtained with nanosheets with crystallinity. After annealing, the derived ZnMgO used for photodetection. It is an interesting paper.

I only have minor comments:

Response: We sincerely thank the reviewer for the positive evaluation and clear summary of our work. We have carefully revised the manuscript to further strengthen the discussion, improve clarity, and include additional relevant references. All modifications are highlighted in blue in the marked-up version of the revised manuscript. Detailed, point-by-point responses to each specific comment are provided below.

(1) Figure 3 N_2 adsorption-desorption isotherm is not well presented. Also the inset is not readable. I suggest that the authors repeat this measurements.

Response: We thank the reviewer for this helpful suggestion. We have re-measured the N_2 adsorption-desorption isotherm and updated Figure 3e to enhance clarity and readability. The revised figure now includes enlarged axis labels and a clearer inset for improved visualization.

In addition, the description in the main text has been expanded to provide more detailed interpretation. The isotherm of ZnMg-MOF-74 exhibits a Type IV profile with an H3 hysteresis loop, indicating slit-shaped mesopores formed by interwoven sheet-like aggregates. This feature corresponds well with the sheet-interlaced morphology observed in SEM (Fig. 2b), confirming the consistency between pore structure and particle arrangement. Revised text can be found on page 6, lines 16 - 22 of the main manuscript. These updates improve both the presentation and the interpretive clarity of the BET analysis.

(2) More discussion on TGA is required. The current discussion may not be quite accurate.

Response: We appreciate the reviewer's insightful comment regarding the need for a more accurate and comprehensive interpretation of the TGA data. In the revised manuscript, we have expanded the discussion of Fig. 3f to clarify the distinct thermal behaviors observed under N_2 and air atmospheres. Specifically, we emphasize that direct annealing in air leads to rapid oxidative decomposition of the organic linkers and subsequent collapse of the framework, whereas annealing under N_2 produces a more stable metal-carbon intermediate. This complementary behavior highlights the rationale behind the two-step annealing process employed in this study—an initial inert pre-anneal that preserves the morphology, followed by an oxidative anneal that removes residual carbon and yields crystalline ZnMgO. In addition, we have cited recent reports on MOF-derived materials utilizing similar two-step annealing strategies under controlled atmospheres to support this interpretation. These clarifications enhance the accuracy of the discussion and establish a clearer

link between the TGA data and the designed thermal treatment pathway. (Revised manuscript, page 7, lines 5 - 11 ; Reference [31])

(3) *The photodetection responses reasons presented are not the most convincing for Fig 6 a to c differences may not be the once included in the paper. Maybe other suggestions are required for the differences.*

Response: We sincerely thank the reviewer for this valuable suggestion. We agree that our original discussion of Fig. 6a-c mainly emphasized the effect of grain size and crystallinity, and thus was not sufficiently comprehensive. In the revised manuscript, we have expanded this section by incorporating the photoluminescence (PL) analysis (Fig. 4d), which provides direct evidence of oxygen-related defect evolution under different annealing conditions. Specifically, the blue-shifted and stronger near-band-edge (NBE) emission observed at 450 °C indicates a higher free-carrier density, consistent with the enhanced responsivity and slower decay time. In contrast, the red-shifted and weaker NBE emission at 550 °C reflects defect compensation or partial healing of oxygen vacancies, resulting in a lower carrier density and faster recombination. The concurrent weakening of defect-related visible emission further supports this interpretation. These results demonstrate that the photoresponse variations arise not only from microstructural changes but also from the evolution of oxygen-related defect states, in good agreement with the PL observations. Revised text can be found on page 8, lines 22 - 27, and page 11, lines 16 – 23 of the main manuscript.

(4) *A final discussion on whether other alloys and strategies can help to expand the concept (use ref Science 389, 372, 2024).*

Response: We sincerely thank the reviewer for highlighting this important point. We fully agree that the broader applicability of the liquid-metal-assisted electrosynthesis concept should be clarified. This valuable suggestion has helped us strengthen the impact and general relevance of our study. In the revised manuscript, we have substantially expanded the concluding section of the *Results and Discussion* to explicitly address how this concept can be extended beyond the Zn-Mg system. Guided by the recent perspective on the “atomic intelligence” of liquid metals (*Science* 385, 372-373, 2024), we now emphasize that the dynamic and reconfigurable active sites of Ga-based liquids provide a versatile platform for selective ion release, uniform atomic dispersion, and stable interfacial control—capabilities that are not attainable with conventional solid electrodes.

This expanded discussion not only directly addresses the reviewer’s suggestion but also situates our work within the emerging framework of liquid-metal “atomic intelligence,” outlining a clearer roadmap for applying this interface-mediated strategy to diverse alloy systems and future device-oriented applications. The revised text can be found on page 12, lines 8 - 17 of the main manuscript (Reference [35]).

(5) *Please expand the results and discussion parts.*

Response: We thank the reviewer for this constructive suggestion. In response, we have added a closing

subsection in the Results and Discussion entitled “Scope and Generalizability of the Interface-Controlled Route.” This new section (i) contextualizes our findings within the emerging understanding of dynamic Ga-based interfaces (Ref. 35), (ii) summarizes the key design principles that enable the extension of this approach to other alloy systems—particularly emphasizing electrochemical selectivity and coordination compatibility, and (iii) outlines potential alloy–ligand combinations for future multimetal MOF synthesis.

These additions strengthen the overall discussion and provide a clearer conceptual framework for extending the liquid-metal interface strategy beyond the present Zn/Mg system. The revised text can be found on page 12, lines 8 - 22 of the main manuscript.

Response to Reviewer 2:

The topic of this manuscript, liquid metals, MOFs, and electrochemical synthesis, is of significant interest to the materials science and engineering community.

For publication in Nature Communications, the novelty aspect of the manuscript needs to be more convincing.

Response: We sincerely thank the reviewer for acknowledging the importance of our research topic and for the constructive feedback on emphasizing the novelty of our work. In the revised manuscript, we have substantially strengthened the discussion of conceptual innovation and technical advancement. New references and expanded discussions have been incorporated to more clearly differentiate our approach from existing studies and to highlight its unique contribution to the field. All modifications are highlighted in blue in the marked-up version of the revised manuscript. Detailed, point-by-point responses to each specific comment are provided below.

(1) Since bimetallic MOFs have been synthesized previously, the authors should explicitly contrast their approach with existing methods. How does their method provide advantages in structural control, stability, or functionality compared with other reported strategies?

Response: We greatly appreciate this important comment. In the revised manuscript, we have expanded both the Introduction and the Results and Discussion sections to explicitly contrast our approach with previously reported strategies and to highlight the distinct advantages of our method.

In the Introduction, we now clarify that conventional electrochemical routes suffer from intrinsic limitations. Specifically, anodic dissolution is typically restricted to single-metal systems and is hindered by surface passivation when active metals such as Mg or Al are used, whereas cathodic deposition often results in compositional inhomogeneity in multimetallic systems and depends strongly on solid substrates, thereby limiting practical applicability. These factors make the electrochemical synthesis of bimetallic MOFs particularly challenging (page 2, lines 4 - 11).

In contrast, our liquid-metal interface strategy utilizes a Mg-Ga alloy anode that enables dynamic and sustained ion release without passivation, allowing controlled dual-metal incorporation under ambient conditions and without substrate dependence (page 3, lines 15 - 25). During electrosynthesis, the liquid Mg-

Ga anode maintains stable ion release—as evidenced by consistent time-current profiles under constant potential—thereby mitigating the well-known instability of active-metal anodes and ensuring reproducible bimetallic assembly. Furthermore, the functionality of the resulting bimetallic MOF is demonstrated through its ZnMgO derivative, which exhibits a fast and reversible UV photoresponse. As summarized in Supplementary Table 1, reports of MOF-derived bimetallic oxides used in photodetection remain scarce, underscoring the significance of the present approach.

In the Results and Discussion, we have also added a comparative overview supported by Supplementary Table 1, summarizing representative bimetallic MOF syntheses reported in recent years. Most prior studies rely on solvothermal or ion-exchange routes performed at elevated temperatures over long reaction times, and they often encounter substrate dependence or compositional nonuniformity. In contrast, our interface-controlled strategy achieves rapid bimetallic assembly at room temperature, in a substrate-free environment, and with tunable stoichiometry (page 3, lines 33 - 38).

These revisions establish a clearer contrast with existing methods and explicitly demonstrate how our strategy provides advantages in structural control, stability, and functionality, thereby reinforcing the novelty and broader impact of our work.

(2) The use of surface-tension-driven flow to regulate Mg ion release is elegant. However, if Mg ions are released at the liquid metal-solution interface, one would expect a concentration gradient extending into the solution. Did the authors probe this experimentally, e.g., through spatially resolved spectroscopy or compositional mapping? Were changes in MgZnO composition or morphology observed as a function of distance from the liquid metal surface? Please make this point clearer in the manuscript.

Response: We are grateful to the reviewer for this thoughtful comment. We realized that our earlier description did not clearly specify where the concentration gradient forms. In the revised manuscript, we now explicitly clarify that the “concentration gradient” refers to an internal (intra-alloy) Mg gradient within the Mg-Ga liquid anode, which is established under bias by electromigration and electrocapillarity. Together with Ga’s inherently high surface tension, this internal gradient drives a surface-tension-gradient (Marangoni) flow at the interface, continuously releasing Mg²⁺ from the alloy surface into the electrolyte. We have clarified that this process represents interface-controlled ion release from the liquid metal rather than a persistent Mg²⁺ gradient extending through the bulk electrolyte between the working and counter electrodes (page 3, lines 15 - 25).

Moreover, recent studies on Ga-based liquid-metal systems have experimentally confirmed that electrochemically induced surface-tension gradients generate interfacial convective flows, such as ring-shaped or vortex-like patterns at the alloy-electrolyte boundary (Adv. Funct. Mater. 32, 2108673 (2022); Reference [26]). These Marangoni-type flows act as an intrinsic dynamic mixing mechanism that continuously refreshes the liquid-metal surface and disperses the released species into the surrounding electrolyte. Such self-induced convection effectively homogenizes the local ionic environment, thereby minimizing any macroscopic

concentration gradient across the cell and enabling the system to operate as a self-sustaining reactor with stable ionic distribution.

Simulations of interfacial phenomena taking place on the liquid alloy upon applying a voltage. (Adv. Funct. Mater. 32, 2108673 (2022))

To make this point clearer, we have implemented the following revisions in the manuscript:

- i. Revised the description in Fig. 1b to specify the presence of an “internal concentration gradient within the liquid metal” and to indicate that Mg²⁺ ions are continuously transported into solution through interfacial Marangoni convection;
- ii. Added new references documenting electrocapillarity- and Marangoni-driven interfacial flow in Ga-based systems, as well as the use of liquid metals as dynamic ion sources; and
- iii. Included compositional mapping of the annealed MOF-derived powders (Supplementary Fig. 8) to confirm that Zn and Mg are uniformly distributed at the particle scale in the final products. Although this analysis pertains to the oxide derivatives rather than the in-situ MOF growth stage, the observed compositional uniformity supports the interpretation that metal ions were homogeneously mixed during the initial coordination process, without localized deposition near the liquid-metal surface.

These clarifications and additional references enhance the clarity of the discussion and address the reviewer’s questions regarding the location of the concentration gradient and the spatial uniformity of the resulting materials.

(3) The use of equations 1 and 2 to explain nucleation and crystal growth does not add new understanding in electrosynthesis of the bimetallic MOF crystal. Are MgO and ZnO synthesized separately or co-dependently? What is the critical nucleus size of this system?

Response: We sincerely thank the reviewer for raising this insightful point, which helped us recognize that our original description may have caused confusion. In the initial submission, Equations (1) and (2) were intended to describe grain-size evolution during thermal conversion of the MOF-derived oxides, but this

distinction was not made sufficiently clear. This might have given the impression that these equations were used to explain MOF assembly itself. To eliminate this ambiguity, we have revised the manuscript to explicitly present the coordination reactions governing the formation of ZnMg-MOF-74 (new Eqs. 1 - 4). In addition, we have replaced the terms “nucleation/growth” with “coordination/assembly” when referring to MOF formation, which more accurately reflects the underlying chemistry (page 3, lines 6 - 14).

For the oxide products, we now clarify that ZnO and MgO crystallize independently during the two-step annealing process. This conclusion is supported by Raman spectroscopy and a newly added schematic (*Supplementary Fig. 7*), which illustrates the sequential transformation pathway: under N₂ atmosphere, bond cleavage and partial carbonization produce a Zn/Mg-C intermediate, whereas under air, the carbon matrix is removed and Zn and Mg separately crystallize into ZnO and MgO domains (page 8, lines 22 - 27). TEM images (Fig. 5) further confirm the presence of distinct crystalline regions separated by amorphous boundaries, while EDS mapping (*Supplementary Fig. 8*) verifies that Zn and Mg remain homogeneously distributed at the particle scale (page 9, lines 8 - 14).

Regarding the reviewer’s question on critical nucleus size, we respectfully acknowledge that determining a precise value would require in situ techniques, which are beyond the scope of this study. Nonetheless, we used classical nucleation theory (Eqs. 1 and 2) as a qualitative framework to interpret the temperature-dependent evolution of grain size. At moderate N₂ pretreatment temperatures (450 °C), the system provides an optimal balance between nucleation and growth, allowing sufficient atomic mobility for stable crystal formation. However, when the temperature is too high (550 °C), rapid thermolysis in nitrogen produces highly dispersed oxide precursors and limits local aggregation of metal atoms required for stable nucleation, resulting in smaller average grains after subsequent oxidation. This explanation aligns with our experimental observation that the grain size follows a small → large → small trend with increasing pretreatment temperature (Fig. 5).

We believe that these revisions—including clarified terminology, the introduction of explicit coordination reactions, and the addition of *Supplementary Fig. 7*—significantly improve the clarity and consistency of the discussion. We again thank the reviewer for this constructive comment, which has substantially strengthened the manuscript.

(4) Figure 5 presents TEM data showing distinct ZnO and MgO crystalline phases but does not provide information on their spatial distribution. How uniform is the MgZnO structure overall? This is also related to the above question. Are the MgO and ZnO domains intimately intermixed, or do they exist as heterogeneous, segregated grains? This distinction is important, as microstructural uniformity would strongly influence potential applications (e.g., in catalysis, sensing, or electronics). High-resolution compositional mapping could help address this point.

Response: We sincerely thank the reviewer for this valuable comment and for highlighting the importance of spatial compositional uniformity. We recognize that our initial submission did not clearly illustrate this aspect. To address this point, we have now included corresponding TEM-EDS elemental mapping (*Supplementary*

Fig. 8) to provide a clearer representation of the Zn and Mg spatial distribution within the oxide structure. The new mapping data confirm that Zn and Mg are uniformly distributed across the ZnMgO particles at the micrometer scale, demonstrating overall compositional homogeneity (page 9, lines 8 - 14).

At higher magnification, TEM images (Fig. 5) reveal that ZnO and MgO crystallize as localized domains separated by amorphous regions rather than forming a fully intermixed lattice. This microstructural configuration is consistent with the two-step annealing mechanism discussed earlier (Supplementary Fig. 7), in which Zn/Mg-C intermediates formed under N₂ are subsequently oxidized in air to yield distinct oxide grains (page 8, lines 22 - 27).

To prevent potential misunderstanding, we have further clarified in the main text that “ZnMgO” refers to a mixed ZnO-MgO oxide derived from the ZnMg-MOF-74 precursor, rather than a single-phase compound. This distinction underscores that the material exhibits macroscopic compositional uniformity while retaining nanoscale structural heterogeneity, a feature that may play a key role in its optoelectronic behavior.

The addition of EDS mapping and clearer terminology clarifies the spatial distribution and structural interpretation of the ZnMgO material, directly addressing the reviewer’s concern regarding compositional uniformity and microstructural coherence.

(5) There is a missed opportunity to provide in-depth understanding of the bimetallic MOF microstructure and nucleation mechanism in contrast to single metal MOF nucleation and crystal growth mechanisms.

Response: We thank the reviewer for this thoughtful suggestion. We realized that our original text did not clearly situate our contribution within the broader evolution from monometallic to bimetallic MOFs, nor did it explicitly explain why we describe MOF formation in terms of coordination/assembly rather than classical nucleation/growth.

To address this, we have revised the Introduction to clearly explain why bimetallic MOFs (BMOFs) have attracted increasing attention. They provide enhanced structural stability, diversified active sites, and tunable electronic and catalytic functionalities. At the same time, we emphasize that their synthesis remains challenging because differences in ionic radii, valence states, coordination geometries, and metal-ligand bond strengths often cause structural heterogeneity or phase segregation. This revision provides clearer context for the motivation behind our approach (page 2, lines 4 - 11).

In the Results and Discussion (Fig. 1a paragraph), we now explain that the key mechanistic advance of this work lies in the use of a liquid-metal anode that continuously releases Mg²⁺ ions while Zn²⁺ ions are supplied from the electrolyte. This configuration enables synchronized dual-metal coordination within a single framework. Unlike conventional anodic deposition, where solid anodes are gradually covered by insulating MOF layers that restrict further metal-ion release, the liquid-metal interface in our system remains conductive and self-renewing. Moreover, because the reaction proceeds in suspension rather than on a fixed substrate, the product forms on the counter electrode (cathodic side) instead of directly on the anode surface. This substrate-

free configuration effectively avoids the passivation and spatial-separation issues that typically hinder the co-deposition of two metal species in conventional setups, enabling stable co-assembly of Zn^{2+} and Mg^{2+} into a uniform bimetallic framework (page 3, lines 6 - 25).

Together, these revisions better highlight the mechanistic insight and conceptual advancement of our interface-controlled strategy, addressing the reviewer's concern and clarifying how this work bridges the gap between single- and bimetallic MOF formation.

Thanks again for the reviewer's very sincere and constructive comments.

We sincerely appreciate your time for the manuscript. Please do not hesitate to contact us should you need any additional information. Thank you very much.

Yours sincerely,

Chun-Wei Huang

Associate Professor

Department of Materials Science and Engineering

Feng Chia University

Tel: 886-4-24517250 ext 5303

E-mail: huangcw@fcu.edu.tw